# Male Infertility in the XXI Century: Are Obesogens to Blame?

**DOI:** 10.3390/ijms23063046

**Published:** 2022-03-11

**Authors:** Ana C. A. Sousa, Marco G. Alves, Pedro F. Oliveira, Branca M. Silva, Luís Rato

**Affiliations:** 1Department of Biology, School of Science and Technology, University of Évora, 7006-554 Évora, Portugal; acsousa@uevora.pt; 2Comprehensive Health Research Centre (CHRC), University of Évora, 7002–554 Évora, Portugal; 3Department of Anatomy, Unit for Multidisciplinary Research in Biomedicine (UMIB), Institute of Biomedical Sciences Abel Salazar (ICBAS), University of Porto, 4050-523 Porto, Portugal; alvesmarc@gmail.com; 4Biotechnology of Animal and Human Reproduction (TechnoSperm), Unit of Cell Biology, Department of Biology, Institute of Food and Agricultural Technology, University of Girona, ES-17003 Girona, Spain; 5Department of Chemistry, QOPNA & LAQV, University of Aveiro, 3810-193 Aveiro, Portugal; pfobox@gmail.com; 6Faculty of Health Sciences, University of Beira Interior, Rua Marquês d’Ávila e Bolama, 6200-001 Covilhã, Portugal; branca@fcsaude.ubi.pt; 7Health School of the Polytechnic Institute of Guarda, 6300-035 Guarda, Portugal

**Keywords:** endocrine disruptors, male fertility, obesogens, adverse outcomes pathways, testicular metabolism, Sertoli cells

## Abstract

The permanent exposure to environmental contaminants promoting weight gain (i.e., obesogens) has raised serious health concerns. Evidence suggests that obesogens are one of the leading causes of the marked decline in male fertility and are key players in shaping future health outcomes, not only for those who are directly exposed to them, but also for upcoming generations. It has been hypothesized that obesogens affect male fertility. By using an interdisciplinary strategy, combining in silico, in vitro, in vivo and epidemiological findings, this review aims to contribute to the biological understanding of the molecular transformations induced by obesogens that are the basis of male infertility. Such understanding is shaped by the use of Adverse Outcomes Pathways, a new approach that may shift the paradigm of reproductive toxicology, contributing to the improvement of the diagnosis and management of the adverse effects of obesogens in male fertility.

## 1. Introduction

The exposure to an increasing number of environmental contaminants has become a prevailing narrative in developed and developing countries. Most of these contaminants are from anthropogenic origin, and some of them are able to interact with the hormonal system, leading to its partial or total disruption. The Endocrine Society has defined these contaminants as endocrine-disrupting chemicals (EDCs) [1]. EDCs are of concern since they have been associated with the development of several diseases, especially those related to metabolism. In the early 2000s, Bruce Blumberg hypothesized that exposure to certain EDCs, particularly tributyltin (TBT), predisposes individuals to weight gain through altered lipid metabolism and increased number of adipocytes. Those EDCs are known as obesogens [2], and the up-to-date science on this topic allowed these contaminants to be considered key players in the obesity’s epidemic. Exposure to obesogenic environments can result in adult-onset diseases, and the evidence gathered over the last few years has demonstrated that humans are more prone to become obese in these high-risk “environments” [3]. As such, people should be aware of the consequences of the permanent exposure to obesogens, especially during critical periods, such as embryogenesis, fetal development, and perinatal life, when organ systems are developing. Although the dominant argument remains regarding healthier daily habits, humans have adopted a negligent behavior, where the regular use of products containing obesogens increases exposure.

When the narrative is about environmental contaminants, people usually question about the acute effects resulting from exposure to high levels of pollutants, but the number of individuals permanently exposed to low doses is higher than the number of individuals experiencing high levels. So, our focus should be on the silent effects caused by the continuous exposure to low levels of obesogens. The United States Environmental Protection Agency (USEPA) defined the no-observed-adverse-effect level (NOAEL), which denotes the maximum level at which there are no biologically adverse effects. Nevertheless, some obesogens present agonist or antagonist activities at concentrations lower than the apparent NOAEL, thus raising serious questions as to what levels we may be safely exposed. The recognition of an association between the continuous toxic effects and the dose to which we are exposed has led to the concept of subclinical toxicity [4]. In other words, the chronic exposure to low levels of obesogenic-disrupting chemicals induces subtle alterations that “may escape” the conventional screening and testing methods for evaluating EDCs [5]. Furthermore, EDCs generally do not follow the traditional toxicological dose–response model, in which higher doses correspond to more pronounced effects, and tend to exhibit non-monotonic dose–response curves [6]. As such, significant efforts have been put forward by the scientific community and by regulatory agencies in order to develop a new framework to evaluate EDCs toxicity. In fact, the Organization for Economic Co-operation and Development (OECD) developed test guidelines for evaluating chemicals for endocrine disruption and promoted an approach to identify the primary cellular targets of toxic substances, together with the subsequent cellular and biochemical responses that eventually lead to adversity [7]. These guidelines were recently updated and the use of adverse outcomes pathways (AOPs) were also included [8]. The AOPs approach, by allowing to establish the links between an endocrine mechanism and its potential apical effects [8], is particularly relevant for male fertility because it is highly vulnerable to the adverse effects of EDCs. In fact, exposure to EDCs has been considered as a possible cause for the increased incidence of idiopathic infertility in modern societies.

Male reproductive health problems resulting from obesogenic toxicity have been under increasing scrutiny, and while some works associated the decline of sperm quality to obesogens’ exposure, others have deepened into the molecular mechanisms on how these contaminants affect male fertility. Obesogens target the male reproductive axis affecting the hormonal signals that control the male reproductive function, especially the complex network of hormones governing testicular cells’ metabolism. Most of those substances are lipophilic, mimicking naturally endogenous hormones. So, obesogens interfering with hormone receptors will compromise genomic and non-genomic mechanisms with adverse consequences for male fertility. Male reproductive function and overall energy metabolism are intimately related, and any dysregulation induced by obesogens in these tightly regulated mechanisms hampers male fertility.

Obesogens also exert other effects independent of the reproductive axis; however, those effects remain elusive. One of the most relevant mechanisms controlling males’ fertility potential is the metabolic cooperation among testicular cells that can be severely affected by obesogens, highlighting new possibilities for therapeutic intervention. The relevance of testicular cell metabolism has gained interest, not only of basic researchers, but also of clinical professionals due to its significance to male fertility. The link between metabolism and reproductive function is exceptionally susceptible to disruptions promoted by environmental contaminants. So, there is an urgent need to unveil the toxic-disrupting mechanisms of obesogens, especially the metabolic crosstalk between Sertoli cells (SCs) and germ cells, since any dysregulation will end up in male infertility.

It has been hypothesized that the toxic-disrupting effects of obesogens affect testicular metabolic cooperation and this may be the solving key to deciphering the information flowing through the environmental–somatic–germ line axis. From a clinical standpoint, the identification of possible mechanisms or biomarkers, as reported recently [9], will give a new impetus in the search for a strategy to improve the diagnostic of male infertility associated with the exposure to environmental contaminants. The existing models of the study have allowed us to understand the toxic effects of obesogens, but the inherent limitations are still an obstacle to a full perception of how these contaminants act in the testicular environment. Most of the studies are accomplished under in vitro or ex vivo conditions, and in this regard, Sertoli cells are suitable models for toxicology studies. SCs are responsible for the physical and nutritional support of spermatogenesis, but due to their distinctive features, these cells are also used as models for a wide range of studies. Recurring efforts have been devoted to developing testicular cells or tissue cultures for basic and clinical research, which are a valuable tool for testing the effects of natural or synthetic agents on testicular tissue [10]. In vivo studies should be used subsequently, for those obesogens that might require further investigation. Studies with animal models, particularly mammals, allow a more robust analysis of the effects of environmental obesogens in humans; still, they present several methodological and ethical constrains that may pose obstacles to their use, data interpretation and ultimately may hamper the conclusions.

Thus, comprehensive studies that link clinical evidence with data on the levels of obesogens in human matrices are highly desirable in order to confirm (or not) the in silico, in vitro, and in vivo results. In this review, we discuss the significant advances in the putative effects of obesogen exposure on male fertility and future challenges in this field. We put forward new approaches focused on metabolism with possible translational potential that might allow mitigating the adverse impact of obesogens in male reproductive health.

## 2. A Brief Overview of Testicular Metabolism

Spermatogenesis is a cellular event of high-energy demand since germ cells must be constantly nourished to meet their energy requirements. This occurs in the seminiferous tubules where the somatic SCs and the germ cells establish an essential metabolic cooperation. This metabolic cooperation is vital for male fertility and depends on the proper functioning of several metabolic pathways. These pathways are under the strict control of a complex network of peripheral hormones and endogenous factors. This is a vulnerable spot where obesogens can modulate male reproduction [11,12].

In metabolic terms, SCs are quite peculiar because these cells do not use sugar as the primary substrate for energy production; instead, they convert the majority of it into lactate that is then delivered to the developing germ cells, especially those located beyond the blood–testis barrier (BTB). The final product of glycolysis is pyruvate, which (1) can be converted into alanine, by alanine aminotransferase; (2) can be converted into acetyl-CoA in mitochondria and enter the Krebs cycle; or (3) can be reduced to lactate by lactate dehydrogenase (LDH), with the- concomitant oxidation of nicotinamide adenine dinucleotide (NADH) into NAD^+^. Once produced, lactate is preferentially exported through monocarboxylate transporter 1 and monocarboxylate transporter 4 to the existing space between SCs and germ cells [13]. Not all germ cells metabolize the same substrates, but those located in the immune privileged site show a lower glycolytic capacity and rely on the supply of lactate, as is the case of pachytene spermatocytes and early spermatids [14]. The relevance of lactate was demonstrated when spermatogenesis was recovered in cryptorchidic animals [15] and when germ cell apoptosis was prevented in humans by intratesticular administration of this metabolite [16]. In addition to lactate, SCs also produce other metabolites that are essential for spermatogenesis, such as cholesterol and other lipid precursors, as, for example, acetate. This is a crossroad metabolite esterified to acetyl-CoA either in the cytoplasm or in mitochondrial matrix by acetyl-CoA synthase. Acetate may contribute to the maintenance of the high rate of lipid synthesis and the intense remodeling of the germ cells’ membranes during their development [17]. Lipids are the preferred energetic substrate of mature SCs and these cells usually degrade residual bodies and phagocyte apoptotic germ cells, enabling the recycling of lipids to produce ATP [18]. Depending on the physiological conditions, SCs may use other substrates, such as amino acids [19] and glycogen [20], which help them to maintain the adequate nourishment of germ cells.

Unlike secluded germ cells, sperm becomes independent of SCs’ nourishment after being released into the tubular lumen. Sperm is metabolically versatile because these cells produce energy through glycolysis, but also oxidative phosphorylation. These processes may occur independently or in combination depending on the “surrounding” conditions. Nevertheless, mammalian spermatozoa can use a variety of substrates, allowing energy production independently from mitochondrial activity, thus making fertilization possible under different circumstances.

## 3. Molecular Connections between Obesogen Toxicity and Testicular Metabolic Pathways: From Fundamental to Clinical Evidence

Testicular metabolism is a complex event that is partly mediated by the endocrine system, so any dysregulation induced by obesogens-disrupting chemicals compromises testicular signaling pathways, especially those related to the metabolic cooperation occurring in the seminiferous epithelium [21]. Furthermore, testicular metabolism is a point of control for spermatogenesis, and potential threats to male reproductive health may be encountered through the permanent exposure to environmental factors, which certainly hamper the metabolism of testicular cells and therefore compromise male fertility (Figure 1). Obesogens mimic natural hormones, and the mechanisms by which they act are diverse and include, for example, nuclear receptor binding and epigenetic modifications [22]. Generally, the most studied mechanism of obesogen action is related to peroxisome proliferator-activated receptor gamma (PPARγ) and 9-cis retinoic acid receptor (RXR) action. In fact, the model obesogen tributyltin (TBT) acts through the peroxisome proliferator-activated receptor gamma (PPARγ) and cooperates with 9-cis retinoic acid receptor (RXR) in the promotion of adipogenesis and lipid storage [2,23]; nevertheless, other metabolic pathways might also be altered [21]. So, it is expected that PPARγ-RXR serves as a link between energy metabolism and reproduction, and possibly mediates the toxic effects of obesogens on testicular cells. Testicular metabolism relies on the adequate functioning of the reproductive axis and the signals resulting from this complex system [24]. The metabolic cooperation established between testicular cells is highly dependent on a strict hormonal network, and the metabolism of each cell type is interconnected [21,25,26,27]. PPARγ, for example, regulates the release of gonadotropin, and the absence of this nuclear receptor decreases the peripheral release of the follicle-stimulating hormone in male mice [28]. PPARγ is also expressed in SCs and is crucial in the expression of genes involved in lipid and glycolytic metabolism. It seems that PPARγ controls the lactic fermentation of SCs by promoting key factors in the pathway, such as glucose uptake and the expression of LDH. Recently, our group observed, under ex vivo conditions, that TBT inhibits the nutritional support of spermatogenesis by SCs, thus illustrating novel mechanisms by which obesogens control male fertility [29]. The expression of PPARγ was also confirmed in sperm and plays a pivotal role not only in the control of energy homeostasis of these cells [30], but also ensuring motility and other key events, such as capacitation and acrosome reaction. The inactivation of PPARγ with the antagonist GW9662 blocked sperm capacitation and acrosome reaction through the decrease in acrosin activity and pentose phosphate pathway [30]. Based on this evidence obesogens may negatively impact the metabolic cooperation of testicular cells, but also sperm function. It should be stressed that the effects of obesogens on sperm quality and male fertility are not limited to the interaction with PPARγ, although the PPARγ-RXR heterodimer is on the frontline of the scientific research. In fact, several obesogens, including pesticides and plasticizers (e.g., phthalates, bisphenol-A (BPA) and BPS) [31,32], may also interfere with other nuclear receptors, or may be responsible for the induction of reactive oxygen species, or may induce DNA damage and alterations on DNA methylation pattern, for example [31]. Thus, given the high complexity of testicular metabolism and the different signaling pathways involved, there are innumerous possibilities for obesogens to interfere, which renders their study particularly changeling.

## 4. Evaluation of Obesogen Toxicity in the XXI Century

Traditional toxicity testing is based on the evaluation of short-term acute toxicity, generally performed in laboratory animals using high doses of the contaminant under study. These studies are then complemented with long-term chronic toxicity testing. However, this standard toxicity testing system is costly, time consuming and requires many animals. Furthermore, given the huge amount of chemicals available, it is virtually impossible to evaluate all of them using the whole animal traditional approach. Thus, in the first decade of the XXI century, a new vision for toxicity testing emerged [33]. This new vision aimed to “transform toxicity testing from a system based on whole-animal testing to one founded primarily on in vitro methods that evaluate changes in biologic processes using cells, cell lines, or cellular components, preferably of human origin” [33]. This novel approach uses computational models to extrapolate data obtained by combining the information generated using nontraditional toxicity testing (see next sections—in silico, in vitro and ex vivo models) and high-throughput screening technologies into the process of chemical hazard evaluation. In the process of this paradigm shift, a new approach to understand the mechanisms of toxicity, including EDCs toxicity was proposed: the Adverse Outcomes Pathway (AOP). An AOP can be described as a “conceptual construct that portrays existing knowledge concerning the pathway of causal linkages between a molecular initiating event (MIE) and a final adverse outcome (AO) at a biological level of organization that is relevant to a regulatory decision” [34]. Taking as an example the recently published AOP for the assessment of EDCs/Reproductive toxicants, the MIE is the activation of PPARα and AOs are the malformation of the reproductive tract in males and impaired fertility [35]. Overall, AOPs promote the use of alternative methods increasing the understanding of biological processes and contributing to decrease in the use of animals in toxicological research [35]. In parallel, AOPs allow to confidently extrapolate data measured at low levels of biological organization (often in vitro) to predict the higher level outcomes [35], and thus are a remarkable tool to identify novel biomarkers that upon proper validation can be used in clinical settings. Given the importance of the in silico, in vitro and ex vivo models in the XXI century toxicity, and the current directive for the reduction in animal experimentation or choose alternatives to animals, this review will focus mainly in these models.

### 4.1. In Silico Models

Over the recent years, in silico models have gained increasing interest. These computer based models can be used, for example, to (i) combine large amount of data, generally associated with gene expression analysis, obtained from different studies [36]; and to (ii) study the mode of action of obesogens based on their chemical structures [37]. For example, le Maire and collaborators [38], by using crystal structural analysis, demonstrated that the activation of PPARγ-RXR pathway by the model obesogen TBT is mediated by RXRα and not PPARγ, because TBT has weak agonistic activity towards PPARγ. Interestingly, triphenyltin (TPT), an organotin compound used as a co-biocide with TBT [39], strongly activates PPARγ, as demonstrated by Harada and collaborators [37]. Such differences between TBT and TPT are, most probably, a consequence of the S…Sn bounding interactions in the activation of PPARs [40]. Frontera and Bauzá [40] provided theoretical and experimental evidences of the presence of functional S…Sn tetrel bonds by using protein data bank analysis and theoretical calculations to analyze the impact of S…Sn bounding interactions in the activation mechanism of PPARs by both TBT and TPT. These authors demonstrated for the first time the existence of S…Sn tetrel bonds in biological media, and thus opened new possibilities to study the impact of such bonds in biological processes. Altogether the available in silico studies allow to understand better how given obesogen interacts with nuclear receptors, for example. However, because in silico models lack the complexity of biological media, conclusions on the mechanisms of obesogens action are not possible. Thus, these models are being combined with in vitro ones in order to predict the mode of action of obesogens better [37].

### 4.2. In Vitro and Ex Vivo Models

The assessment of testicular toxicity poses enormous challenges due to the anatomical and physiological complexity of the tissue. For this reason, it is necessary to select the most appropriate model to identify possible cellular targets and disclose the molecular mechanisms that mediate the toxic effects. Cultures of SCs have arisen as the preferred models for evaluating male reproductive toxicity, based on studies with isolated cells from which we can distinguish two different cell culture types: primary cultures and cell lines. The establishment of SCs lines is a valuable tool in attempting to clarify the effects of environmental contaminants on these cells. They have been widely used; however, they present some limitations since they do not present all characteristics of SCs isolated from the native tissue [41,42]. This is why primary cultures of SCs are still considered the best cellular model for toxicological purposes [43]. SCs are vulnerable to contaminants and their adverse effects are primarily associated with structural changes and disruption of BTB dynamics, which compromises the “movement” of germ cells through the epithelium [44]. However, other essential events to spermatogenesis, such as the metabolic control of SCs, are also important and thus have received increasing interest from the scientific and clinical community. In this context, it is necessary to prioritize potential upcoming methods to assess the effects of obesogens in the metabolic performance of the SCs. The metabolism of SCs have been extensively studied by our group [13,17,20,25,45,46] and compelling evidence has demonstrated that obesogens reprogram the metabolism of SCs. Considering its vulnerability to the disrupting effects of obesogens, SCs have been preferably used (see Table 1). Nevertheless, it must be taken into account that intracellular signaling is poorly represented in two-dimensional-cell cultures, as under in vivo conditions SCs present a columnar morphology and a spatial arrangement that certainly conducts to different cellular responses than those observed in monolayers. So, three-dimensional in vitro cultures are more suitable to mimic the specificities of native tissues. Few years ago, a 3D SC–gonocyte co-culture model was developed by Yu and collaborators [47], who succeed to establish a culture model maintaining the structural organization that closely mimics the one observed within seminiferous tubules. However, despite the apparent advantages, these models failed to replicate the real in vivo conditions. Furthermore, both 2D and 3D SC cultures only provided information on the mechanisms occurring in cells, as opposed to happens in vivo [48]. Predicting toxic response in reproductive tissues remains a significant challenge and to overcome the shortcomings of the cellular models, organs cultures maintained under ex vivo might be a valuable option. Recently, Goldstein and collaborators [49] successful maintained the physiological cellular organization and germ cell maturation of isolated seminiferous tubules from prepubertal rats. This system showed to be promising to simulate in vivo histological changes in response to obesogens. Similarly, Roulet and collaborators [10] also developed a system that helps to predict the response of human testicular tissue to obesogen exposure. Those authors were able to preserve several cellular and molecular aspects, such as 3D architecture, cell viability, pathway activity, and global gene expression of small slices of adult human testis. This may be a promising technique allowing to surpass the conflicting results that arise from culture systems based in isolated cells or disintegrated biopsies. The need for basic research, but mainly, for clinical applications led to significant efforts to develop culture systems allowing the study of the impact of environmental factors on the human testicular function. Given the groundbreaking advances in the biotechnology, materials science and engineering fields, it is possible that in the near future more integrated solutions, such as organoids and “organ-on-a-chip” [50,51], will be available to study the impacts of obesogens on testicular function. In fact, a microfluidic device was already developed and the testis tissues cultured in this device maintained spermatogenesis successfully for 6 months [52]. Such systems, together with the in silico, in vitro and ex vivo models will further contribute to reduce the use of the animal models traditionally used in toxicity assays.

## 5. Gaps and Future Challenges

Exposure to EDCs has been considered as a possible leading cause for the increased rates of male infertility in modern societies. This raises serious societal and economic challenges since the costs associated with the treatment of male infertility due to EDC exposure reached almost 15 billion EUR /year in Europe alone [61]. Despite the substantial advances carried out over the past few years, important questions remain to be solved and major challenges need to be tackled before the real impact of obesogens in male fertility can be assessed and preventive measures can be implemented. Such gaps and challenges include, for example: (i) the need to obtain robust epidemiological data on the levels of obesogens and their association with male reproductive outcomes, (ii) the exposure to multiple contaminants and the subsequent cocktail effect, and (iii) the possibility of obesogens to alter the epigenome and consequently to induce effects in future unexposed generations. Most of our current knowledge on the toxic effects of obesogens comes from in vitro studies and from animal models, in which all the confounders are well controlled and the effects of obesogen exposure can be investigated prospectively. However, in humans, prospective studies are rare and most of the available epidemiological data comes from retrospective studies [62,63,64,65,66,67,68,69,70]. It should also be stressed that human studies are subjected to bias and confounders that need to be addressed. It is common, for example, to perform studies in specific groups of the population (e.g., men from infertility clinics) and, as such, the extrapolation to the general population is difficult [71]. Ideally, population-based studies should be performed and the levels of different obesogens evaluated in human samples (e.g., urine for hydrophilic compounds, such as BPA [63], and blood/serum for lipophilic ones, such as persistent organic pollutants [72]), alongside with biomarkers of testicular function. Such biomarkers can be disclosed by using omics technologies and high-throughput screening, in order to better describe the metabolic pathways associated with obesogen exposure. This approach may provide new opportunities and advances in our understanding of the testicular metabolic pathways and how they can be manipulated at the clinical level. It is well established that exposure to multiple EDCs can result in metabolic disturbances at concentrations for which no effect is observed when exposure occurs individually [73,74]. This cocktail effect remains one of the greatest challenges to be addressed in the future. Thus, in order to effectively evaluate the risk associated with exposures to multiple chemicals, there is an urgent need to develop new integrative approaches. Such approaches must combine the evidence gathered from exposomics, clinical and epidemiological data with evidence derived from basic research, including in silico, in vitro, ex vivo and in vivo models. In order to integrate such data, the use of the AOP approach for mixtures is particularly relevant. By using this approach, it is possible to select better the method to be used in mixture risk assessment and it is also possible to use novel approaches for determining chemical groups and, consequently, to understand how to utilize them in a risk assessment context, as recently suggested by Nelms et al. [75]. The AOP approach is also useful when trying to understand the chronic impacts of obesogens, including their multi- and transgenerational toxicity. It is now well established using in vivo models that ancestral exposure to obesogens results in metabolic (e.g., obesity) and epigenetic alterations in future generations [76,77,78]. Reproductive alterations were also described in rats, up to the third generation after being exposed in utero to vinclozolin. The F3 generation (that corresponds to great grandchildren) exhibited increased testicular germ cell apoptosis and decreased sperm motility [79]. The observed changes were associated with changes in DNA methylation in the F3 generation sperm [79]. Recently, it was proposed that the increased incidence of diseases may be associated with transgenerational changes in the epigenome and transcriptome of future generations as a consequence of ancestral exposures to EDCs [80]. However, the mechanism by which this epigenetic transgenerational inheritance occurs is still not completely understood and constitutes a tremendous challenge and an important opportunity. In fact, the knowledge of the epigenetic changes and patterns associated with specific contaminants and with specific diseases will allow to identify epigenetic biomarkers signatures that might be used to evaluate patients’ individual susceptibility and thus facilitate preventive medicine, as recently suggested by Skinner and collaborators [80].

## 6. Conclusions

The link between metabolism and reproductive function is highly susceptible to environmental contaminants that interact with the endocrine system, and today there is no doubt that our lifestyle and the surrounding environment play a vital role. Since the emergence of the obesogen hypothesis in 2006, major efforts have been in place in order to unveil the possible associations between low-dose exposure to obesogens and male infertility and, most importantly, to describe in a mechanistic way the molecular pathways in the testicular metabolism affected by such exposures. Despite the evidence gathered to date, there are still many unknowns and thus further research on this topic is mandatory. Such research should be performed collectively by gathering researchers from different fields, as only with the close collaboration of basic scientists, clinicians, environmental chemists, statisticians, engineers and regulatory scientists will it be possible to collect and analyze real samples and thus, to develop new models and biomarkers that can be used by clinicians and by policymakers to inform legislators on the best actions to implement in order to reduce exposure to obesogens and ultimately to protect human health.

## Figures and Tables

**Figure 1 ijms-23-03046-f001:**
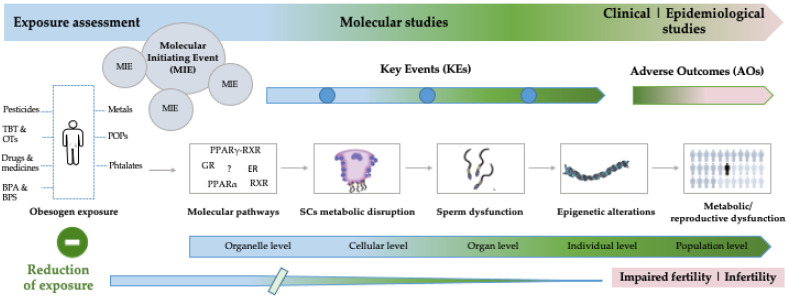
Overview of the impact of obesogens on reproductive function with the indication of the potential sources of obesogens known to affect the male reproductive system, alongside with the depiction of the Adverse Outcome Pathways. BPA: bisphenol-A; BPS: bisphenol-S: ER: estrogen receptor; GR: glucocorticoid receptor; OTs: organotin compounds; POPs: persistent organic pollutants; PPARα: peroxisome proliferator-activated receptor alpha; PPARγ: peroxisome proliferator-activated receptor gamma; RXR: 9-cis retinoic acid receptor; SCs: Sertoli cells; TBT: tributyltin.

**Table 1 ijms-23-03046-t001:** Summary the main obesogens and their proposed effects on metabolism in reproductive cells/tissues of mammals.

Obesogens	Specie(s)/ Tissue(s)/Cells	Doses		Effects on Metabolism
2,4-D	Rat SCs	100 nM, 10 µM, 1 mM		↓GLUT3, PFK1 LDH mRNA, ↓Lactate production [25]
BPA	Rat testis	0.005, 0.5, 50, 500 µg/kg body wg/day		↓IRS-1, ↓GLUT2 [53] ↓HEX, ↓PFK [54]
CPYF	Rat testis	0, 2.7, 5.4, 12.8 mg/kg body wg		↑LDH [55]
Lead	Rat SCs	0.01, 0.05, 0.1 mM		↑Lactate production [56]
PCBs	Rat SCs	10^−7^ M (PCB22) 10^−8^ M (PCB77)		↑Lactate production [57]
PIO	Rat SCs	1, 10, 100 µM		↑Glucose uptake ↓GLUT3 ↑Lactate production ↑LDH ↑MCT4 [58]
PTLs	Rat	Testis	CE-2 diet with 2%(mass) of DEHP	↓ACC ↑LCAD ↑3KACT [59]
		SCs	0.1–200 µM	↑Pyruvate production ↑Lactate production [60]
TBT	Rat SCs	0.1 nM, 10 nM		↓Glucose uptake ↓Pyruvate uptake ↓GLUT1 ↓Lactate production [11]

Legend: Abbreviations: 2,4-D: 2,4-dichlorophenoxyacetic; 3KACT: 3-ketoacyl-CoA thiolase; ACC: Acetyl CoA carboxylase; BPA: Bisphenol A; CPYF: chlorpyrifos; DEHP: bis(2-ethylhexyl) phthalate; GCs: germ cells; GLUT1: glucose transporter 1; GLUT2: glucose transporter 2; GLUT3: glucose transporter 3; HEX: hexokinase; IRS-1: insulin receptor substrate 1; LCAD: long-chain acyl-CoA dehydrogenase; LDH: lactate dehydrogenase; MMP: mitochondrial membrane potential; n. d.: non-determined; PCBs: polychlorinated biphenyls; PFK: phosphofructokinase; PFK1: phosphofructokinase 1; PIO: pioglitazone; PTLs: phthalates; SCs: Sertoli cells; TBT: tributyltin; wg: weight; ↑ increase; ↓ decrease.

## Data Availability

Not applicable.

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
