# Peer review of "Male Infertility in the XXI Century: Are Obesogens to Blame?"

_ijms, 2022, doi:10.3390/ijms23063046_

Round 1

Reviewer 1 Report

Interesting and well-written mini-review. I think that the authors went to the point and give a good contribution to the field.

Author Response

The authors would like to thank this reviewer for the commentaries. We have made some alterations according with the suggestions of reviewer 2. We have address them in the attachment.

Reviewer 2 Report

ijms-1558393:  Male infertility in the XXI century: Are obesogens to blame?

The authors have provided a review of the current scientific literature associating links between increased exposures to environmental EDCs that act as obesogens, and the decrease in male fertility observed over the past many years.  They cover the clinical and epidemiological literatures, and well as some of the more mechanistic – especially metabolic and endocrine – research that has been conducted over the past several years examining different aspects of this link.

Included in in their discussion is a strong presentation of some of the methodological strengths and weaknesses of different conceptual frameworks and methodologies used to address the role of obesogen exposure and changes in male fertility. They make a strong argument for the importance of using an Adverse Outcomes Pathways approach that integrates data from multiple methodologies and levels of analysis.

All-in-all, this is a strong review which will make a solid contribution to the scientific literature.

Comments for editing the manuscript:

  • I really had difficulty following Figure 1, a diagram which was intended to simplify the sources of relevant obesogens and the possible mechanisms by which they affect male fertility.
    • Why are pesticides listed twice as sources?
    • What is the relationship of the parts of the diagram surrounding the figure of the man and sources (as the title suggests). If this is really a second, more mechanistic overview of the literature, that should be stated, perhaps changing the title.
    • More importantly, I had trouble following the various outcomes (infertility treatment, reduction of exposure, impaired fertility, reduction of exposure) and the various physiological responses.

I think this figure should be redrawn and clarified.

  • Smaller suggestions:
    • The introductory paragraph (lines 26-115) should be broken down into several smaller paragraphs for greater ease of reading and comprehension of the considerable materials presented.
    • Sentence on lines 121-123 needs to be rewritten. As written, the pathways are secreted by germ cells.  Rewrite for clarity.
    • Line 178 needs editing. “Release of gonadotropin release” should be reworded.
    • Lines 312-318 seem redundant of the previous lines, where the points about decreased use of in vivo animal models was clearly stated.

Obviously, except for the figure, these are minor edits.

Author Response

Dear Reviewer,

the authors would like to thank you for your constructive commentaries. We have altered the manuscript accordingly, and we hope we have improved it. All changes were addressed in the letter to reviewers and were marked as track-changes in the file.

Thank you very much.

Reviewer 3 Report

There is a well-designed and written paper. In my opinion, the topic is extremely important, in some aspects new and necessary. The authors are aware of strengths and limitations od this study.

Good job!

Author Response

Dear Reviewer, 

the authors thank you for your appreciation of the manuscript. We have made some changes according to the reviewer#2 suggestions. 

Thank you very much.
